# A Pedagogical Introduction to the Lifshitz Regime †

**Robert D. Pisarski** [1,*,‡] **, Vladimir V. Skokov** [2,3,‡] **and Alexei Tsvelik** [4,‡]

1    Department of Physics, Brookhaven National Laboratory, Upton, NY 11973, USA
2    RIKEN/BNL Research Center, Brookhaven National Laboratory, Upton, NY 11973, USA; vladi@skokov.net
3    Department of Physics, North Carolina State University, Raleigh, NC 27695, USA
4    Condensed Matter Physics and Materials Science Division, Brookhaven National Laboratory, Upton, NY 11973, USA; tsvelik@bnl.gov
*    Correspondence: rob.pisarski@gmail.com
†    This paper is based on the talk at the 7th International Conference on New Frontiers in Physics (ICNFP 2018), Crete, Greece, 4–12 July 2018.
‡    These authors contributed equally to this work.

**Abstract:** We give an elementary and pedagogical review of the phase diagrams which are possible in quantum chromodynamics (QCD). Herein, emphasis is upon the appearance of a critical endpoint, where disordered and ordered phases meet. In many models, though, a Lifshitz point also arises. At a Lifshitz point, three phases meet: disordered, ordered, and one in which spatially inhomogeneous phases arise. At the level of mean field theory, the appearance of a Lifshitz point does not dramatically affect the phase diagram. We argue, however, that fluctuations about the Lifshitz point are very strong in the infrared and significantly alter the phase diagram. We discuss at length the analogy to inhomogeneous polymers, where the Lifshitz regime produces a bicontinuous microemulsion. We briefly mention the possible relevance to the phase diagram of QCD.

**Keywords:** QCD; phase diagram; critical endpoint; Lifshitz; spatially ordered

## 1. Introduction

Experiments at the Relativistic Heavy Ion Collider (RHIC) and the Large Hadron Collider (LHC) have demonstrated conclusively that a new state of matter is produced in the collisions of heavy ions at high energy. In the central region, the system behaves like a quark–gluon plasma (QGP) at high temperature and very small baryon chemical potential.

The central region was studied originally because it is (almost) free of baryons. This is useful because, in thermodynamic equilibrium, it is possible to form comparisons with the results of numerical simulations on the lattice.

It is natural to ask what happens as the system goes down in energy. In that case, even in the central region, the temperature decreases, and the baryon (or quark) chemical potential becomes significant. In the collisions of heavy ions, it will never be possible to reach a very low temperature, but, clearly, the phase diagram as a function of temperature $T$ and quark chemical potential $\mu$ is probed.

At low $\mu$ and nonzero $T$, numerical simulations on the lattice indicate that while there is no true phase transition, there is a large increase in pressure in a relatively narrow region of temperature [1]. That is, there is a crossover that appears to be associated with the chiral transition.

This need not remain true as the chemical potential increases. It is plausible that, at increasing $\mu$, a line of first-order transitions arises. If so, the line of first-order transitions must end in a critical endpoint [2–4]. This is a true critical point, so some correlation lengths are infinite in thermodynamic equilibrium.

In this paper, we discuss what appears to be a minor feature in the phase diagram: the appearance of spatially inhomogeneous phases. In nuclear matter, these are the familiar pion and kaon condensates. The appearance of such phases is difficult to derive, even in mean field theory. Nevertheless, although these phases are certainly important, naively, one would not expect such condensates to dramatically affect the phase diagram.

This is true at the level of mean field theory. We show, however, that fluctuations dramatically affect the phase diagram [5]. In mean field theory, three phases meet at what is known as a Lifshitz point. In three spatial dimensions, the fluctuations at a Lifshitz point are so strong that they completely wipe out the Lifshitz point, leaving only a Lifshitz regime. It is possible that the critical endpoint is completely wiped out, leaving only a line of first-order transitions. In this case, while infrared fluctuations can be strong in the infrared, they remain finite at all points in the phase diagram.

While our arguments are qualitative, they are rather general. We also discuss the close analogies between the phase diagram of quantum chromodynamics (QCD) and that of inhomogeneous polymers [6,7]. In this regard, what we call the Lifshitz regime is known as bicontinuous microemulsions and is of practical importance.

## 2. Mean Field Theory: Tricritical Points

We first review the standard theory of how a critical endpoint can arise. Consider a scalar field $\phi$, which we assume for simplicity to be single component. It is trivial and immediate to the case where $\phi$ transforms under some global symmetry group $\mathcal{G}$. We take the following as the Lagrangian:

$$\mathcal{L} = \frac{1}{2} \left( \partial_i \phi \right)^2 + \frac{1}{2} m^2 \phi^2 + \frac{1}{4} \lambda \phi^4 + \frac{1}{6} \kappa \phi^6 \ . \tag{1}$$

If we consider the behavior at nonzero temperature in four space–time dimensions, then static correlation functions are determined by correlation functions in three spatial dimensions. In that case, $\phi$ has dimensions of $\sqrt{}$ mass, so $\lambda$ has dimensions of mass, while $\kappa$ is dimensionless. Thus, in the sense of the renormalization group, $\kappa$ is a marginal operator and should be included.

Let us begin with the case where $\lambda$ is positive. Then, we have the standard phase diagram. The theory is invariant under a global symmetry of $Z(2)$, $\phi \to -\phi$. When $m^2$ is positive, the expectation value $\langle \phi \rangle = 0$, and one is in the symmetric phase. For negative $m^2$, $\langle \phi \rangle \neq 0$, which is the broken phase. There is a second-order phase transition when $m^2 = 0$. By the renormalization group, the behavior is controlled by the universality class of a $Z(2)$ invariant theory, such as the Ising model. For other models, the universality class is that of the symmetry group $\mathcal{G}$.

It is also possible to consider negative quartic couplings, where $\lambda < 0$. To ensure that the potential is bounded from below, we have to assume that the hexatic coupling $\kappa$ is positive. Then, one has a first-order transition from the symmetric to the broken phase. It is possible to determine in detail where a transition from $\langle \phi \rangle = 0$ to some nonzero $\phi_0$ occurs. By the overall $Z(2)$ symmetry, there are two vacua, with $\pm \phi_0$. This is determined by the potential being degenerate with $\phi = 0$, so $V(\pm \phi_0) = 0$; for $\phi_0$ to be a minimum, $\partial V(\phi) / \partial \phi = 0$ at $\phi_0$. There are two conditions, which can be satisfied for a given value of $\lambda < 0$ by adjusting $m^2$. The basic point can be understood without detailed computation: $m^2$ must be positive. For example, at $m^2 = 0$, the potential about the origin decreases, and so the value of the potential at $\phi = 0$ is always above that at $\phi_0 \neq 0$.

This gives the phase diagram in Figure 1. There is a line of second-order transitions for $m^2 = 0$ and $\lambda > 0$ and a line of first-order transitions when $m^2 > 0$ and $\lambda < 0$. They meet at the origin, where $m^2 = \lambda = 0$. This is a tricritical point, where both the mass and quartic coupling vanish. In three dimensions, the hexatic coupling runs logarithmically in the infrared, as it is the upper critical dimension for this interaction.

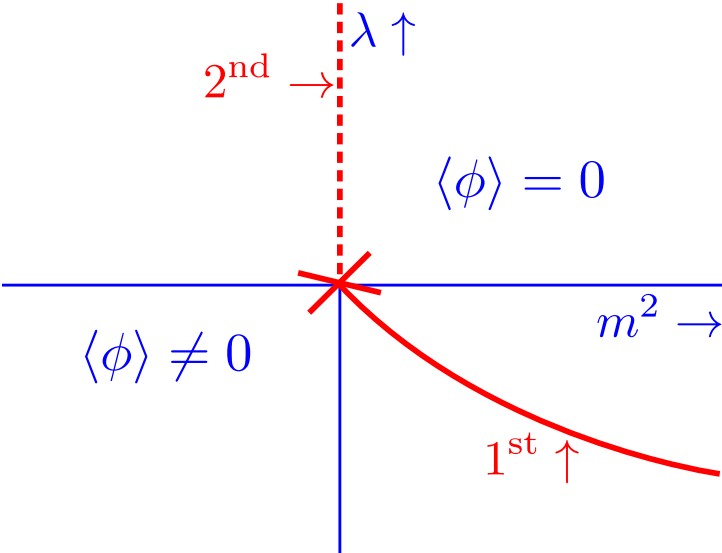

**Figure 1.** The standard diagram in mean field theory, with a tricritical point at the origin.

If the symmetry is not exact, one adds a term which breaks the symmetry, such as $h\phi$. If the background field $h$ is small, the position of the transitions should be near that for $h = 0$. The most dramatic change is that the line of second-order transitions becomes a line for crossover. That is, when $h \neq 0$, the field always has a nonzero expectation value, $\langle \phi \rangle \neq 0$, so the theory is always in a broken phase.

For sufficiently negative values of $\lambda$, though, the first-order transition must remain. Thus, the line of first-order transitions persists. This implies that it terminates in a critical endpoint, precisely as it does for the liquid–gas transition in water. The critical field is $\phi - \langle \phi \rangle$. If the underlying theory has a larger symmetry of $\mathcal{G}$, the universality class of the critical endpoint remains as in the Ising model, $Z(2)$. This is because the field develops an expectation value along some direction, and the critical field is still $\phi - \langle \phi \rangle$ for that particular direction.

### 3. Mean Field Theory: Lifshitz Point

Next, we consider a more general Lagrangian,

$$\mathcal{L} = \frac{1}{2}\left(\partial_0 \phi\right)^2 + \frac{1}{2M^2}\left(\partial_i^2 \phi\right)^2 + \frac{Z}{2}\left(\partial_i \phi\right)^2 + \frac{1}{2}m^2\phi^2 + \frac{1}{4}\lambda\phi^4 \ . \tag{2}$$

This is an effective Lagrangian, so it is possible to have terms involving higher derivatives, at least for terms with spatial derivatives. To respect causality, any terms with time derivatives must be of second order. We then include a term with four spatial derivatives, $\sim (\partial_i^2 \phi)^2$; by dimensionality, this coefficient must have dimensions of $\sim 1/M^2$, where $M$ is some mass scale which arises by constructing the effective theory. To ensure the theory has a stable vacuum, this coefficient must be positive. This implies that the usual term, with two spatial derivatives, can have a coefficient, $Z$, which is negative.

Consider first the case where $Z$ is negative in the symmetric phase, with $m^2 > 0$. This dispersion relation is plotted in Figure 2. There is no condensate, but, clearly, the minimum of the propagator is at a nonzero momentum, $k_0$. We can choose this direction to be along, for example, $k_z$. Expanding $\vec{k} = (k_0 + k_z, k_\perp)$, we require that the terms $\sim k_z k_0$ vanish. The inverse propagator is then

$$\frac{1}{M^2}(\vec{k}^2)^2 + Z\vec{k}^2 + m^2 = m_{\text{eff}}^2 - 2Z(k_z - k_0)^2 + \frac{1}{M^2}\left(4k_0 k_z \vec{k}^2 + (\vec{k}^2)^2\right) \ , \quad Z < 0 \ . \tag{3}$$

where

$$k_0^2 = - Z \frac{M^2}{2} \; , \; m_{\text{eff}}^2 = m^2 - \frac{Z^2}{4} M^2 \; . \tag{4}$$

The first condition is only satisfied if $Z$ is negative. As can be seen from the effective mass, having $Z < 0$ tends to drive the effective mass to the negative, but if $m^2$ is sufficiently large, we can still remain in the symmetric phase.

It is notable that in Equation (3), the terms that are quadratic in the transverse momenta, $k_\perp^2$, vanish identically. This is due to the spontaneous breaking of the rotational symmetry: the propagator has a minimum about some nontrivial value, and we choose a direction about which to expand. This is also why there are terms $\sim k_z \vec{k}^2$ in the inverse propagator.

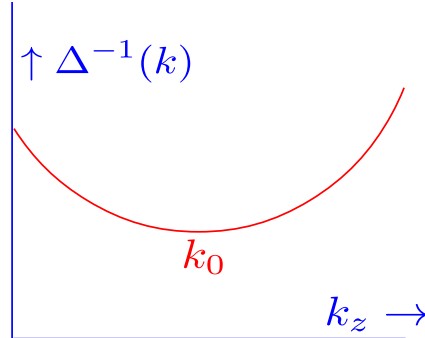

**Figure 2.** The dispersion relation for the Lagrangian of Equation (2) in the symmetric phase, where $Z < 0$ and $m^2 > 0$.

As $Z$ becomes more negative, we are eventually driven into a phase when the symmetry is broken locally, with $\langle \phi \rangle \neq 0$. To lowest order, the kinetic term is negative, though this is a qualitatively different state. In detail, the nature of this state intricately depends upon the symmetry group. We chose to discuss the very simplest possibility, where $\phi$ now has two components. In that case, we assume that, along some arbitrary direction, which we choose to be $\hat{z}$, there is a spiral:

$$\phi(x) = \phi_0(\cos(k_0 z), \sin(k_0 z)) \; . \tag{5}$$

We then have two parameters to determine, $k_0$ and $\phi_0$. The kinetic terms contribute

$$\frac{1}{2} \left( \frac{k_0^4}{M^2} + Z k_0^2 \right) \phi_0^2 \; . \tag{6}$$

Minimizing with respect to $k_0$ gives

$$k_0^2 = - \frac{Z}{2} M^2 \; . \tag{7}$$

When $Z < 0$, this is the lowest energy state, with $k_0 \neq 0$. Using this value for $k_0$, the value of the condensate is determined by the usual equation,

$$V(\phi) = \frac{1}{2} m_{\text{eff}}^2 \phi^2 + \frac{\lambda}{4} (\phi^2)^2 \; . \tag{8}$$

Minimizing this potential gives the usual value for the condensate,

$$\phi_0^2 = - \frac{m_{\text{eff}}^2}{\lambda} \; . \tag{9}$$

In this spatially homogeneous phase, while $\langle \phi \rangle \neq 0$ locally, it is not the case globally. This is obvious even for one condensate oriented in a given direction, because when we integrate over $z$, $\langle \phi(z) \rangle$ will clearly average to zero.

Further, this state is itself unstable, as we show later. Even without computation, this can be guessed. There is nothing special about the $\hat{z}$ direction, and fluctuations will tend to disrupt the theory. It is natural to expect that there is instead a series of patches, whose width is determined by the underlying dynamics of the theory. We discuss this later.

Without going into the details, we can understand the nature of the phase diagram in mean field theory, which we illustrate in Figure 3. If $Z$ is positive, then we have the usual second-order transition from the symmetric to a broken phase when the mass squared vanishes. Consider $Z < 0$. From the form of the effective mass in Equation (4), it vanishes when $m^2 = ZM^2/4$. Thus, we expect a second-order phase transition as we cross this line, as indicated in the figure.

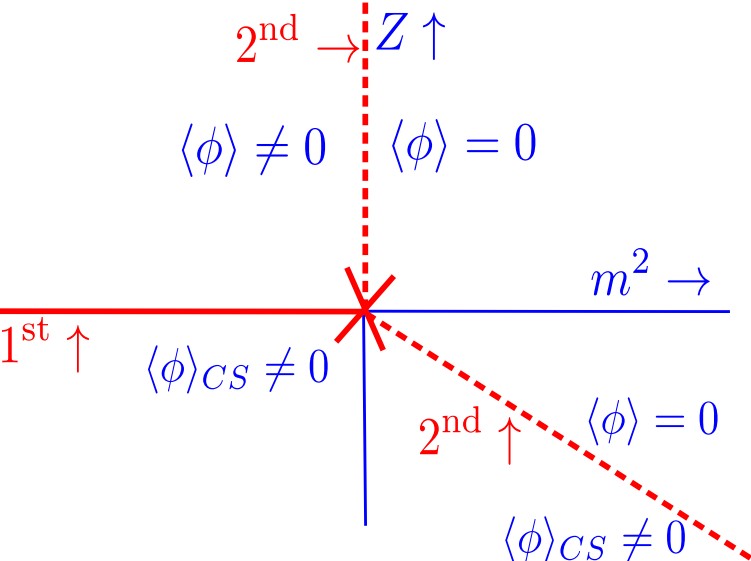

**Figure 3.** The phase diagram in mean field theory for a theory in which $Z$ can be of either sign. There are three phases: symmetric, with $\langle \phi \rangle = 0$; broken, with $\langle \phi \rangle \neq 0$; and a spatially inhomogeneous phase, denoted by CS for chiral spiral. The three phases meet at the Lifshitz point, where $m^2 = Z = 0$.

When $m^2$ is negative, one goes from a broken phase to one which is spatially inhomogeneous as $Z$ becomes negative. At the level of mean field theory, the free energy is given by

$$V(\phi_0) = -\frac{m_{\text{eff}}^4}{4\lambda} \,. \tag{10}$$

Assume that $Z$ is linear in temperature, $Z = a(T - T_c)$, about the critical temperature $T_c$. By Equation (4), there are then terms linear in $T - T_c$ in the potential and therefore the free energy. This is the sign of a first-order transition, as then the derivative of the free energy with respect to temperature is discontinuous.

This is also obvious physically. In a typical first-order transition, the theory jumps from one phase to another, and the masses are discontinuous. In this case, the masses are continuous, but the structure of the theory is completely different as one goes from a homogeneous ground state to a ground state dominated by patches of spatially inhomogeneous condensates.

## 4. Anisotropic Fluctuations and the Phase Diagram

The phase diagram changes dramatically once fluctuations are included. The basic physics can be understood from the propagator in the symmetric phase (Equation (3)). Because the minimum is at

nonzero momentum, the ground state spontaneously breaks Lorentz symmetry, and the fluctuations are anisotropic. To one-loop order, there is a contribution to the mass term,

$$\Delta m^2 \sim \lambda \int d^2 k_\perp dk_z \frac{1}{(k_z - k_0)^2 + m_{\text{eff}}^2 + \dots} \sim \lambda \int^M d^2 k_\perp \int_{m_{\text{eff}}} dk_z \frac{1}{(k_z - k_0)^2} \sim \lambda \frac{M^2}{m_{\text{eff}}} . \quad (11)$$

For small effective masses, the dominant contribution is from $k_z - k_0 \sim m_{\text{eff}}$, and the anisotropic propagator makes the theory effectively one-dimensional. This is the origin of the term $\sim M^2/m_{\text{eff}}$. The integral over transverse fluctuations, $k_\perp$, is cut off by the higher-order terms in the propagator, proportional to the mass scale associated with the higher derivative terms, $\sim M$.

The effective reduction to one dimension produces the factor of $1/m_{\text{eff}}$. This implies that while in mean field theory there is a second-order transition as $m_{\text{eff}} \to 0$, this is not consistent with fluctuations. This does not preclude a phase transition from occurring: for a fixed negative value of $Z$, one is clearly in a symmetric phase for large positive $m^2$ and in a spatially inhomogeneous phase for negative $m^2$. Thus, a phase transition must happen, but it will do so through a first-order transition, jumping from one nonzero value of $m_{\text{eff}}^2$ to another.

In condensed matter physics, this was first pointed out by Brazovski [8,9]. In this context, it is often referred to as a fluctuation-induced first-order transition, but it is rather different from, for example, the Coleman–Weinberg phenomenon. The latter only arises for theories with more than one coupling constant: under renormalization group flow, one of the coupling constants flows into negative values, thereby triggering a first-order transition. This flow is a detailed function of both the dimensionality of space–time and the symmetry group under which the fields transform.

The present case is very different: whatever the original dimensionality of space–time, because of the negative kinetic term, the infrared fluctuations are those of an essentially one-dimensional theory. Similarly, the symmetry under which the fields transform is irrelevant, since all that matters is that the fluctuations are one-dimensional. Of course, the details of the transformation do depend intimately upon these factors. However, the basic point is simply that, at low momentum, the theory is one-dimensional, and it is not consistent to have an interacting, massless theory in one dimension.

This implies that the line of second-order transitions that separates the symmetric and spatially inhomogeneous phases is, in fact, a line of first-order transitions. This produces the phase diagram in Figure 4.

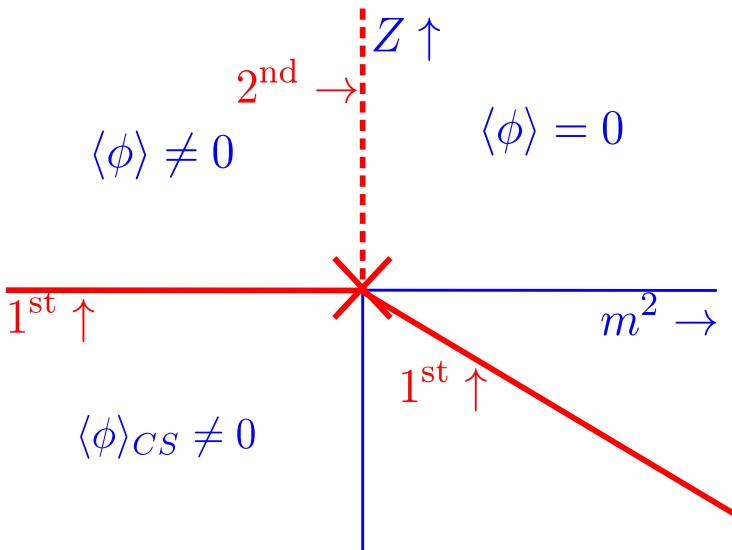

**Figure 4.** The phase diagram for a theory where $Z$ can be of either sign, corrected for anisotropic fluctuations. The three phases meet at the Lifshitz point.

The anisotropic form of the propagator is similar in the phase with spatial anisotropy. Assume that, in the standard broken phase, the theory breaks to a subgroup $\mathcal{H}$ of the original group $\mathcal{G}$. If $U$ denotes the Goldstone bosons of $\mathcal{G}/\mathcal{H}$, the effective Lagrangian in the spatially inhomogeneous phase is

$$
\begin{aligned}
\mathcal{L} &= f_\pi^2 \mathrm{tr}\,|(\partial_z - i\,k_0)U|^2 + \frac{c_1}{M}\,\mathrm{tr}\left((\partial_z - i\,k_0)U^\dagger \partial_\perp^2 U + \text{c.c.}\right) \\
&+ \frac{c_2}{M^2}\,\mathrm{tr}|\partial_\perp^2 U|^2 + \frac{c_3}{M^2}\,\mathrm{tr}\left(\partial_\perp^2 U^\dagger(\partial_z - i\,k_0)^2 U + \text{c.c.}\right) + \dots .
\end{aligned}
\tag{12}
$$

This is the natural generalization of the propagator in the symmetric phase (Equation (3)). The coefficients $c_i$ are to be determined. The important point is that, to quadratic order in the fluctuations, only the longitudinal momenta enter.

Consider again a tadpole diagram in this phase. For simplicity, we set $c_1 = c_3 = 0$,

$$
\int d^2 k_\perp \int dk_z \, \frac{1}{(k_z - k_0)^2 + (k_\perp^2)^2/M^2} \sim \int d^2 k_\perp \, \frac{M}{k_\perp^2} \sim \log(\Lambda_{IR}) ,
\tag{13}
$$

where $\Lambda_{IR}$ is an infrared cutoff. Because of this divergence, there is no true long-range order. This is exactly analogous to the smectic-C phase of liquid crystals: these are systems which are ordered in one direction but act as a liquid in the transverse directions [10].

The lack of long-range order is to be expected. After all, we had assumed that the theory had broken the three-dimensional to one-dimensional symmetry. As commented, we expect that the theory forms patches, with one-dimensional ordering within each patch. The interaction between the patches is controlled by the logarithmic infrared divergences above. Nevertheless, there can be a large separation of scales, so for large Fermi spheres, there can be many patches.

## 5. Isotropic Fluctuations

We now consider the effect of fluctuations near the Lifshitz point. At the Lifshitz point, the effective theory is given by

$$
\mathcal{L} = \frac{1}{2}(\partial_0 \phi)^2 + \frac{1}{2M^2}\left(\partial^2 \phi\right)^2 + \frac{\lambda}{4}\phi^4 .
\tag{14}
$$

Although it will not enter into our considerations at nonzero temperature, which is governed by static correlation functions, we stress that the time derivative is customary of quadratic order.

In momentum space, the propagator at the Lifshitz point is

$$
\Delta(k) = \frac{1}{(\vec{k}^2)^2} .
\tag{15}
$$

We can now use the standard renormalization group analysis of phase transitions. The upper critical dimension is eight when the renormalization of the coupling constant,

$$
\Delta\lambda \sim -\lambda^2 \int d^8 k \, \frac{1}{(\vec{k}^2)^2((\vec{p} - \vec{k})^2)^2} ,
\tag{16}
$$

develops a logarithmic divergence. Similarly, the lower critical dimension is four when the shift in the mass squared,

$$
\Delta m^2 \sim -\lambda \int d^4 k \, \frac{1}{(\vec{k}^2)^2} ,
\tag{17}
$$

is logarithmically divergent. This implies that, in *fewer* than four dimensions, there are power-like infrared divergences, and it cannot be possible to reach the Lifshitz point.

For field theories with an ordinary propagator, $\sim 1/k^2$, as is well known, the upper critical dimension is four, and the lower critical dimension is two. The latter is familiar: it is not possible to have interacting massless modes in two or fewer dimensions.

This implies that it is not possible to reach a Lifshitz point in four or fewer spatial dimensions. The question is then how the phase diagram in Figure 4 is modified by the effects of fluctuations. For a theory with an exact global symmetry, there must still be the phase boundaries indicated, with a second-order phase transition between the symmetric and broken phases and a line of first-order transitions between the spatially inhomogeneous phase and the other two. The only difference is that it is not possible to reach the point where $Z = m^2 = 0$.

It is useful to consider the analogy to a spin system in two (or fewer) dimensions. Begin in the symmetric phase and tune the mass to decrease. Then, unlike in more than two dimensions, it is not possible to tune the mass to vanish: a nonzero mass will be generated non-perturbatively.

For a theory near the Lifshitz point, there are now two parameters to consider, $Z$ and $m^2$. Consider moving along the line of second-order phase transitions. The mass squared must vanish along this line. Further, this line of second-order transitions must meet the line of first order. Consider the endpoint of this line of second-order transitions. The simplest possibility is that $m^2$ vanishes at this endpoint. This implies that, even if $Z = 0$ in mean field theory, a nonzero value of $Z$ is generated *non*-perturbatively. If so, then the universality class of the critical endpoint is the same as along the critical line.

Conversely, in mean field theory, it is only possible to go to a spatially inhomogeneous phase when $Z$ is negative. It is possible, however, that due to strong non-perturbative fluctuations, the theory develops a spatially inhomogeneous phase even if $Z$ is positive but small.

That is, to avoid the instability of a Lifshitz point, everywhere along the line of first-order transitions, either $Z$ or $m^2$ is nonzero. It is possible to have an isolated point where $Z = 0$, but then $m^2$ *must* be nonzero. We suggest the following: there is a region, which we term the "Lifshitz regime", where the values of $Z$ and $m^2$ are dominated by non-perturbative fluctuations. The possible phase diagram is illustrated in Figure 5.

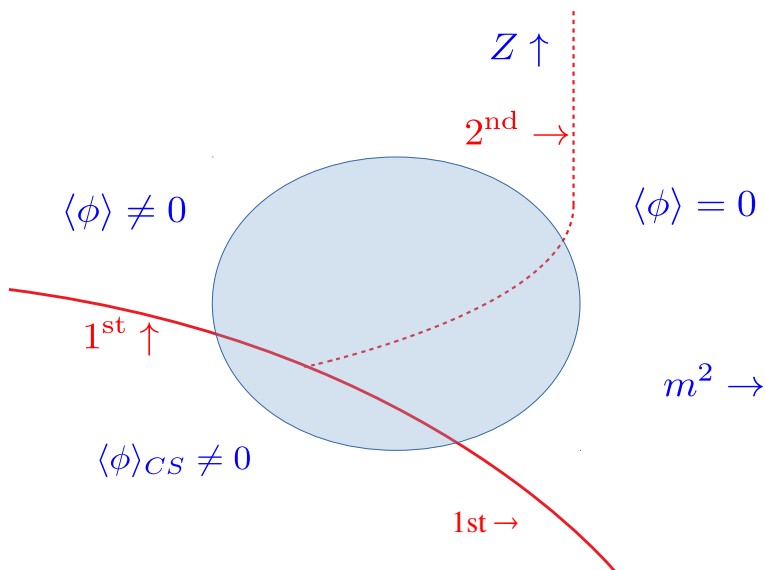

**Figure 5.** The proposed phase diagram, including all fluctuations. The axes $Z$ and $m$ are not indicated, because the Lifshitz point, with $Z = m^2 = 0$, is not accessible. The shaded region is the Lifshitz regime, where $Z$ and/or $m^2$ are generated non-perturbatively due to strong infrared fluctuations.

## 6. Lifshitz Regime in Inhomogeneous Polymers

There is a known example of a would-be Lifshitz point in inhomogeneous polymers [6,7,11–13]. Consider first the example of a mixture of oil and water, which separate into droplets of either oil or water. By adding a surfactant, however, the interface tension between the phases changes, and other phases emerge. A more controlled example is given by mixing two different types of polymers, formed of monomers of type A and of type B, which also separate. In addition, one can also add A-B diblock copolymers, which are long sequences of type A polymer followed by long sequences of type B.

These A-B copolymers localize at the boundaries between phases with only A or B polymers: the part with type A sticks into the part with type A, and similarly for type B. The result is that adding the A-B copolymer decreases the surface tension between the A and B phases.

There are three possible phases. At very high temperature, A and B polymers mix uniformly, which is a symmetric phase. At low temperature, a mixture of A and B polymers separate into regions with only A or B homopolymers and is a broken phase. By adding the A-B copolymer, one can obtain a lamellar phase, where A and B regions form stripes. This is like a smectic liquid crystal, albeit without orientational order.

Mean field theory predicts the existence of a Lifshitz point where these three phases meet. In contrast, both experiments and numerical simulations with self-consistent field theory indicate that there is *no* Lifshitz point [6,7,11–13]: see, e.g, Figure 3 of Reference [6].

Instead, a new, intermediate region emerges near where the Lifshitz point was expected, and it is termed a bicontinuous microemulsion. In this region, the surface tension is essentially zero. The theory forms a spongelike structure with large entropy, where the polymers exhibit nearly isotropic fluctuations in composition with large amplitude.

In terms of an effective theory, the surface tension is proportional to the wave function renormalization, $Z$. Thus, the bicontinuous microemulsion is a region where $Z$ is very small and $m^2$ is nonzero. This is what we call the Lifshitz regime.

## 7. Relation to QCD

We conclude by briefly discussing the possible relevance to QCD. In general, there are two possible instabilities. One is that the quartic coupling constant, $\lambda$, turns negative and generates a critical endpoint. The second possibility is the wave function renormalization constant for the quadratic spatial derivatives, $Z$, becomes negative and generates a Lifshitz regime.

At present, the relationship between the two can only be studied by using effective models. In the simplest Nambu–Jona-Lasino (NJL) model, the two points coincide. This can be understood as the following. We start with

$$\mathcal{L}_{NJL} = \bar{q}\, i \slashed{\partial}\, q + (\bar{q}q)^2 \ . \tag{18}$$

Bosonizing this by introducing $\sigma = \bar{q}q$, at one-loop order, we need to evaluate

$$\mathrm{tr}\log(\slashed{\partial} + \sigma) \approx d_1 \left( \sigma^4 + (\partial\sigma)^2 \right) + \dots , \tag{19}$$

for some constant $d_1$. We only indicate the first terms in this expansion, as there is an infinite series of terms involving higher powers of derivatives and factors of $\sigma$.

What is found, however, is that the coefficients of the first two terms are tied together. While surprising at first, this can be understood through a simple scaling argument: we can rescale both length, $\slashed{\partial} \to \kappa\, \slashed{\partial}$, and $\sigma \to \kappa\sigma$. The one-loop determinant is invariant under this scaling, so any expansion must respect it as well.

The first coefficient, $\sigma^4$, controls the location at which the quartic coupling becomes negative. The second coefficient, $(\partial\sigma)^2$, determines when $Z$ becomes negative. This explains why the critical endpoint and the Lifshitz point coincide in the simplest NJL model. See, for example, Figure 6 of Buballa and Carignano [14]. This is also seen in solutions of Schwinger–Dyson equations [15].

This equality fails when more complicated models are considered. While the critical endpoint and Lifshitz points coincide in the simplest NJL model, this occurs because then the sigma mass is exactly twice the constituent quark mass, $m_\sigma/m_{\mathrm{qk}} = 2$. Carignano, Buballa, and Schaefer [16] showed that, in a quark-meson model, where one can allow $m_\sigma/m_{\mathrm{qk}} \neq 2$, the Lifshitz and critical endpoints separate.

In the plane of temperature $T$ and quark (or baryon) chemical potential $\mu$, as $T$ decreases, the first singularity which one meets is either the critical endpoint or the (would-be) Lifshitz point. Since one can only use effective models, clearly a definitive answer cannot be given.

This suggests the following scenario. As in the NJL and related models, the critical endpoint and the would-be Lifshitz point—which becomes, in fact, a regime—are close to one another. The critical endpoint is dominated by a single massless mode, which exhibits true infrared fluctuations in the infinite volume limit. The Lifshitz point has fluctuations which are large, but finite, in the infinite volume limit.

For the case of heavy ion collisions, which occur over a finite region of space and time, it is clearly a challenge to distinguish between the two types of infrared fluctuations. This is not as difficult as it may seem. For the critical point, there is a single massless mode due to the $\sigma$ meson,

$$\Delta_\sigma(k) = \frac{1}{k^2} . \tag{20}$$

For the Lifshitz regime, not only the $\sigma$ but also pions and kaons exhibit a modified dispersion relation, in which the usual quadratic propagator becomes

$$\Delta_{\text{Lifshitz}}(k) = \frac{1}{(k^2)^2/m^2 + m_{\text{eff}}^2} . \tag{21}$$

In principle, it should be possible to distinguish between the truly massless mode of a critical endpoint from a modified dispersion relation in the Lifshitz regime. Certainly, the latter will exhibit characteristic deviations from the usual statistical models. The crucial test will be in measuring fluctuations: in the Lifshitz regime, those of kaons may be as large as those of pions. Both scenarios will affect the fluctuations in the net proton number.

A proposed phase diagram for QCD is given in Figure 6. The example of Nambu–Jona-Lasino (NJL) models shows that the possibility of a Lifshitz point cannot be ignored. Because of strong fluctuations, the Lifshitz point becomes a Lifshitz regime and will significantly impact any analysis of a (possible) critical endpoint.

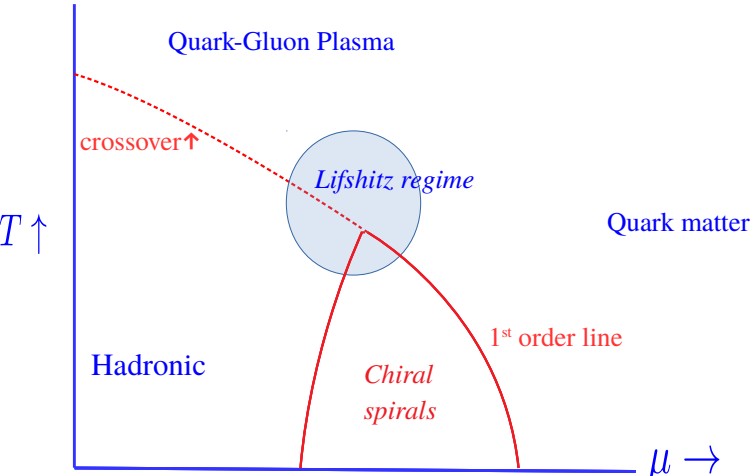

**Figure 6.** A proposed phase diagram for quantum chromodynamics (QCD) in the plane to temperature $T$ and chemical potential $\mu$. We assume that the critical endpoint lies in the region of spatially inhomogeneous phases or chiral spirals. There is an unbroken line of first-order transitions which encloses the regime with chiral spirals.

**Author Contributions:** All authors contributed equally to this research.

**Funding:** R.D.P. is funded by the U.S. Department of Energy for support under contract DE-SC0012704; A.M.T. is funded by Condensed Matter Physics and Materials Science Division, under the the U.S. Department of Energy, contract No. DE-SC0012704.

**Conflicts of Interest:** The authors declare no conflict of interest.

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
