# Peer review of "A Pedagogical Introduction to the Lifshitz Regime"

_universe, doi:10.3390/universe5020048_

Reviewer 1 Report

The manuscript presents a pedagogical review of the Lifshitz regime. It is clearly written, however there are a few minor typos/corrections that the authors should consider before it can be published.

Page 1, line 30: “In this paper we discuss one appears” -> “In this paper we discuss what appears”

Page 2, line 52: please re-formulate the sentence “ There is a second order phase transition for m^2 =  0, which is a second order phase transition.”

Page 4, 1st line: “ The dispersion relation is plotted in Fig. Figure 2 .” -> “ The dispersion relation is plotted in Fig. 2.”

Page 4, line before Eq. (5): “that there is a spiral:” -> “there is a spiral:”

Page 6, line 128: “one of the coupling constants flow” -> “ one of the coupling constants flows”

Page 7, line 141: “Assume that in the standard broken phase, that the theory” -> “Assume that, in the standard broken phase, the theory”

Page 8, line 156: “This implies that in less  than four dimensions, that there are” -> “This implies that, in less  than four dimensions, there are”

Page 8, line 173-174: “That  implies that even if in mean field theory Z =  0, that Z…” -> “ That implies that, even if in mean field theory Z =  0, Z…”

Page 8, lines 176-177: “ It is possible, however, that due to strong non-perturbative fluctuations, that the  theory develops” -> “ It is possible, however, that due to strong non-perturbative fluctuations, the theory develops”

Page 9, lines 188-189: Please re-formulate the sentence “ To this A-B diblock copolymers, which are long sequences of type A polymer, followed by type B.”

Page 10, line 215: “term” -> “terms”

Page 10, line 224: The sentence “In the simplest NJL model,” is not finished.

Page 10, line 227: “ that the Lifshitz and critical endpoints separate” -> “the Lifshitz and critical endpoints separate”

Page 11, line 238: “it’s” -> “its”

Author Response

We thank the referee for their comments.  We have changed all errors of wording pointed out, and going through the manuscript again to make sure that there were none that were missed.

Reviewer 2 Report

In this mansucript the role of the Lifshitz regime in phase diagrams is explained. The title says it is a pedagogical introduction which describes the style of the manuscript very well. Starting from well known cases the main idea is carefully developed and explained. At the end the connection to QCD is made.

The only thing I can criticize is that some sentences were written sloppily and there are, for example, missing words (e.g. This the lowest energy state with), or doubled phrases (e.g., There is a second order phase transition for m2 = 0, which is a second order phase transition.). I saw such sentences in lines 30, 52, 58, 188, 215, 224-225, 232-233 and in the first line below Fig. 2, the first line below eq. (7), the first line below eq. (16). I am not sure if the journal corrects such things automatically before publication.

In addition, the following points caught my attention:
Does p_0 below eq. (5) mean k_0 from eq. (5)?
In Figs. 3 and 4 it says that the phase diagram for a theory with Z<0 is="" shown.="" this="" confusing="" for="" since="" in="" the="" plots="" z="">0 is shown as well.
Does the asterisk in Fig. 6 refer to the critical point? If so, should it not be in the region of chiral spirals as mentioned in the caption?

Author Response

We thank the referee for their comments.  We have changed all of the errors of wording which the referee pointed out.  In addition, we have gone over the manuscript once again, and hopefully caught all others.  Lastly, in the final figure we deleted T_0.  This refers to the point at which T is maximum, with respect to mu.  It is a point of equal concentration.  However, this is a technical point which didn't seem worth emphasizing.